# Novel Lactotransferrin-Derived Antimicrobial Peptide LF-1 Inhibits the Cariogenic Virulence Factors of *Streptococcus mutans*

**DOI:** 10.3390/antibiotics12030563

**Published:** 2023-03-13

**Authors:** Junyuan Luo, Zening Feng, Xiaohui Lyu, Linglin Zhang

**Affiliations:** 1Department of Endodontics, Shanghai Ninth People’s Hospital, Shanghai Jiao Tong University School of Medicine, College of Stomatology, Shanghai Jiao Tong University, National Center for Stomatology, National Clinical Research Center for Oral Diseases, Shanghai Key Laboratory of Stomatology, Shanghai Research Institute of Stomatology, No. 639, Zhizaoju Road, Shanghai 200011, China; 2Department of Endodontics, Shanghai Stomatological Hospital, Fudan University, No. 356, East Beijing Road, Shanghai 200001, China; 3State Key Laboratory of Oral Diseases and National Clinical Research Centre for Oral Diseases, Sichuan University, No. 14, Section 3, Renmin Road South, Chengdu 610041, China

**Keywords:** lactotransferrin, antimicrobial peptides, dental caries, *Streptococcus mutans*, cariogenic virulence

## Abstract

We previously developed a novel lactotransferrin-derived antimicrobial peptide, LF-1, with selective antibacterial activity against the characteristic cariogenic bacterium *Streptococcus mutans*. This study further investigated the effects of LF-1 on the cariogenic virulence factors of *S. mutans* and evaluated the changes in virulence-associated enzymes and genes; the viability, acidogenicity, and aciduricity of planktonic *S. mutans*; and initial colonisation and biofilm formation after treatment with LF-1. The method of qRT-PCR was used to evaluate *S. mutans* virulence-associated gene expression. LF-1 interfered with the cell viability of *S. mutans* within 6 h. LF-1 inhibited the acidogenicity and aciduricity of *S. mutans,* with reduced lactic acid production and survival in a lethal acidic environment, and inactivated lactate dehydrogenase and F_1_F_0_-ATPase activity. LF-1 decreased surface-adherent *S. mutans* within 60 min and inhibited *S. mutans* biofilm formation, where scanning electron microscopy and confocal laser scanning microscopy showed reduced extracellular matrix and bacteria. LF-1 downregulates *S. mutans* virulence-associated gene expression. LF-1 inhibited the growth and cariogenic virulence factors of *S. mutans* in vitro with a reduction in key enzymatic activity and downregulation of virulence-associated gene expression. LF-1 has promising application prospects in the fight against *S. mutans* and dental caries.

## 1. Introduction

Dental caries is one of the most prevalent chronic infectious diseases that seriously affect oral and systemic health [1]. Cariogenic bacteria in dental biofilms are critical pathogenic factors for dental caries [2]. Among these cariogenic bacteria, several epidemiological and animal studies have shown a strong correlation between *Streptococcus mutans* and dental caries, highlighting its strong cariogenic features and revealing the vital role of *S. mutans* in the progression of dental caries [3].

*S. mutans* possesses typical cariogenic biological characteristics (namely cariogenic virulence factors), including high acidogenicity, aciduricity, adhesion, and exopolysaccharide (EPS) synthesis [4]. *S. mutans* relies exclusively on glycolysis and metabolises ingested carbohydrates for energy production while inevitably generating large amounts of acidic metabolites [5]. These acidic products accumulate in dental biofilms and acidify the local microenvironment to a critical pH (pH ≤ 5.5), further leading to tooth demineralisation and cavity formation [6]. Therefore, acidogenicity confers *S. mutans* with a representative cariogenic potential to initiate dental caries. This acidic microenvironment is not conducive to bacterial growth and reproduction. To thrive at low pH values, *S. mutans* maintains intra- and extra-cellular buffer balance through an acid tolerance response and avoids acid damage [7,8]. Consequently, aciduricity ensures that *S. mutans* survives and continuously produces acids, which further aggravates the process of dental caries. Bacterial adhesion and extracellular matrix accumulation are responsible for dental biofilm formation [9,10]. *S. mutans* possesses multiple high-affinity surface adhesions that readily enable colonisation of the tooth surface via specific recognition [11]. The colonised *S. mutans* can utilise its glycosyltransferase (GTF) system to decompose ingested sucrose into large amounts of glucans, further leading to the accumulation of the EPS matrix [12,13]. Insoluble EPS serves as a structural scaffold for biofilms and leads to more bacterial adhesion and aggregation. The EPS matrix wrapping abundant bacteria creates a dense and enclosed microenvironment where the acidic metabolites produced cannot be neutralised and buffered in time, thus further promoting local acidification [14]. Therefore, one strategy for caries control is to inhibit these virulence factors [15], which determine the high cariogenic capacity of *S. mutans*.

Antimicrobial peptides (AMPs) have attracted considerable attention in recent years [16]. Considering that AMPs possess some advantages over conventional anticaries drugs, including definite antibacterial activity, less drug resistance, and fewer adverse reactions, they have become a new approach for caries control [17,18]. Among the AMPs utilised in caries research, we successfully developed a novel AMP, LF-1, based on the functional domain of lactotransferrin using a biomimetic strategy. In previous studies, LF-1 displayed potent antibacterial activity against *S. mutans* and its biofilm and induced notable protection in an *S. mutans*-mediated rodent model of dental caries [19]. Compared with oral commensal *Streptococcus* species (*S. sanguinis* and *S. gordonii*), LF-1 selectively targeted *S. mutans* with preferable membrane affinity. These results highlight the potential of LF-1 to fight *S. mutans* and dental caries [20]. However, the influence of LF-1 on cariogenic virulence factors of *S. mutans* remains elusive and requires further investigation. In addition, there have been few studies on the effects of AMPs on the cariogenic virulence factors of *S. mutans*. Our study could enrich the development of anticaries drugs against *S. mutans*.

Therefore, this study aimed to explore the effects of LF-1 on the cariogenic virulence factors of *S. mutans* and evaluate the changes in virulence-associated enzyme and gene transcription levels. We adopted relatively low drug concentrations (≤8 μmol/L) as the treatment concentrations to avoid the strong bactericidal effects of LF-1 [19]. The null hypothesis was that LF-1 does not suppress *S. mutans* cariogenic virulence factors or the expression of key enzymes and virulence-associated genes. Based on our previous research, this study acquired an enhanced discovery, and our results further corroborate LF-1 with satisfactory anticaries activity against *S. mutans*.

## 2. Results

### 2.1. LF-1 Interfered with the Cell Viability of Planktonic S. mutans

The methyl thiazolyl tetrazolium (MTT) assay indicated that low concentrations (<8 μmol/L) of LF-1 inhibited the viability of planktonic *S. mutans* in a 6-h treatment (Figure 1A). When treated with the minimal inhibitory concentration (MIC) and 1/2 MIC (8 and 4 μmol/L) of LF-1, the relative cell viability of *S. mutans* significantly decreased to under 5% compared to the untreated control (*p* < 0.05). Moreover, LF-1 at 1/4 MIC (2 μmol/L) also significantly inhibited the growth of *S. mutans* with a reduced relative cell viability of 58.27 ± 3.61% compared to the untreated control (*p* < 0.05).

### 2.2. LF-1 Inhibited the Acidogenicity and Aciduricity of Planktonic S. mutans

LF-1 affected the acidogenicity of *S. mutans* in a concentration-dependent manner. As shown in Figure 1B, LF-1 at 2, 4, and 8 μmol/L significantly reduced lactic acid production in *S. mutans* to 0.98 ± 0.11, 0.36 ± 0.08, and 0.33 ± 0.04 mmol/L, respectively, compared with the untreated control (6.83 ± 0.10 mmol/L; *p* < 0.05). LF-1 at 2 μmol/L delayed the glycolytic pH drop in the culture medium, the terminal pH of which was significantly higher than that of the untreated control (*p* < 0.05), and the broth pH remained static at 6.5 within 90 min of treatment with LF-1 at 4 and 8 μmol/L (Figure 1C). Furthermore, the lactate dehydrogenase (LDH) activity of *S. mutans* was significantly decreased after treatment with LF-1 (*p* < 0.05; Figure 1D), corroborating the reduced acidogenicity of *S. mutans*.

Analogously, LF-1 affects the aciduricity of *S. mutans* in a lethal acidic environment (pH 5.0). As shown in Figure 1E, LF-1 at 2 μmol/L significantly reduced the survival of *S. mutans* by over 99% to the untreated control (*p* < 0.05), with the increased LF-1 concentrations leading to a further decrease in the number of viable bacteria. LF-1 significantly inhibited F-ATPase activity in a concentration-dependent manner (*p* < 0.05).

### 2.3. LF-1 Decreased S. mutans Initial Colonisation and Biofilm Formation

There was an overall decrease in *S. mutans’* initial colonisation after treatment with LF-1 gradients (Figure 2A). When treated with LF-1 at 2, 4, and 8 μmol/L for 30 min, the surface-attached *S. mutans* were significantly reduced to 66.86 ± 5.69%, 72.75 ± 6.22%, and 24.71 ± 3.63% of the untreated control, respectively (*p* < 0.05; Figure 2A). When the reaction time was increased to 60 min, this inhibition of initial colonisation was more pronounced in a concentration-dependent manner.

The crystal violet (CV) staining assay indicated that 4 and 8 μmol/L of LF-1 significantly inhibited *S. mutans* biofilm formation, where the biofilm biomass of 4 and 8 μmol/L treatment decreased to 69.45 ± 0.29% and 0.48 ± 0.29% of the untreated control, respectively (*p* < 0.05; Figure 2B,C). Following the CV staining results, LF-1 at 4 and 8 μmol/L significantly reduced the synthesis of water-insoluble EPS in the *S. mutans* biofilm (*p* < 0.05; Figure 2D). Furthermore, 2 μmol/L of LF-1 could significantly retard the lactic acid production in the *S. mutans* biofilm to 84.64 ± 1.95% of the untreated control (*p* < 0.05; Figure 2E), and this inhibition of acid production was gradually enhanced with increased LF-1 concentrations.

### 2.4. LF-1 Alters the Ultrastructural Morphology of the S. mutans Biofilm

Scanning electron microscopy (SEM) observations further confirmed the inhibition of *S. mutans* biofilm formation with LF-1 treatment (Figure 2F). No apparent morphological changes were observed in the *S. mutans* biofilm treated with 2 μmol/L of LF-1 compared to the untreated control, displaying a complex biofilm structure formed by a dense extracellular matrix containing aggregated *S. mutans*. Exposure to 4 μmol/L of LF-1 began to change the surface morphology of the *S. mutans* biofilm, where the loose and thin biofilm structure was accompanied by a visually reduced extracellular matrix and bacteria. Furthermore, 8 μmol/L LF-1 dramatically interfered with *S. mutans* biofilm formation, with little bacterial adhesion.

Subsequently, confocal laser scanning microscopy (CLSM) was used to observe LF-1-induced three-dimensional structural changes in *S. mutans* biofilms. As shown in Figure 3A, the labelled biofilm of the untreated control exhibited dense and uniform red fluorescence (EPS) with massive embedded green fluorescence (bacteria). No apparent changes occurred in the red fluorescence after treatment with 2 μmol/L LF-1, whereas the green fluorescence significantly decreased (*p* < 0.05; Figure 3B,C). Furthermore, the red and green fluorescence gradually weakened with increasing treatment concentrations. The red and green fluorescence signals of the 8 mol/L treatment were very faint, at only 9.04 ± 1.40% and 12.78 ± 3.24% of the untreated control, respectively. In addition, quantitative fluorescence analysis revealed that LF-1 significantly reduced the thickness of *S. mutans* biofilms (including EPS and bacteria) in a concentration-dependent manner (*p* < 0.05; Figure 3D,E).

### 2.5. LF-1 Downregulated the S. mutans Virulence-Associated Gene Expression

The expression profiles of S. mutans virulence-associated genes after treatment with LF-1 are shown in Figure 4. LF-1 at 8 μmol/L significantly reduced virulence-associated gene expression (*p* < 0.05). Notably, half of the virulence-associated genes (including *gtfD*, *ldh*, *atpD*, *vicR*, and *comD*) were significantly downregulated by only 2 mol/L LF-1 treatment (*p* < 0.05).

## 3. Discussion

In our previous studies, we successfully developed a novel lactotransferrin-derived AMP, LF-1, and evaluated its antibacterial activity against multiple caries-associated bacterial strains [19]. LF-1 demonstrated the most notable antibacterial activity against the tested strains of *S. mutans*, with an MIC of 8 μmol/L. In addition, the higher membrane affinity of LF-1 for *S. mutans* than other oral commensal *Streptococcus* species contributes to better targeting of *S. mutans* [20]. Considering the selectivity of LF-1 against *S. mutans* and the crucial role of the cariogenic capacity of *S. mutans*, we further explored the effects of LF-1 on the cariogenic virulence factors of *S. mutans* [4]. We compared the main findings of LF-1 previously reported with those obtained in this study, which is summarised in Table 1. Our results revealed that LF-1 thoroughly suppressed all cariogenic virulence factors (including acidogenicity, aciduricity, adhesion, and EPS synthesis), corroborating the potential of LF-1 as a promising anticaries agent against *S. mutans*.

Concentrations of LF-1 greater than 16 μmol/L can trigger strong bactericidal activity against *S. mutans* with rapid killing kinetics [19], which is detrimental to the assessment of changes in cariogenic virulence factors. Therefore, lower concentrations (≤8 μmol/L) of LF-1 were used in this study. *S. mutans* generally enters the mid-logarithmic phase within 6–8 h with the most vigorous growth speed [21,22], which is when *S. mutans* is most suitable for drug effect evaluation. Above all, we utilised MTT (a cell viability detection reagent) to investigate the effects of LF-1 at low concentrations on *S. mutans* growth at 6 h. Consistent with our previous results [20], LF-1 at 8 and 4 μmol/L almost inhibited *S. mutans* growth within 6 h. Unexpectedly, LF-1 at a low concentration (2 μmol/L) also significantly decreased *S. mutans* growth activity at this concentration but did not delay growth in the time-growth kinetic curve. It is convenient to investigate bacterial growth characteristics by measuring the optical density of bacterial suspensions, but the MTT method is more accurate in exploring differences that cannot be reflected in the time-growth kinetic curve. The inhibition of *S. mutans* growth at low concentrations (≤8 μmol/L) of LF-1 suggests its potential to affect the cariogenic virulence factors of *S. mutans*.

Glycolysis is the exclusive pathway of *S. mutans* carbohydrate metabolism that generates acidic by-products [23]. As a typical cariogenic bacterium, *S. mutans* is characterised by its acidogenic capacity [24]. LF-1 significantly suppressed the lactic acid production of *S. mutans* and retarded the rate of pH decline in the culture medium, revealing an impairment in the acidogenicity of *S. mutans*. LDH is a key rate-limiting enzyme in the glycolysis of *S. mutans* and plays a crucial role in determining the acidogenicity of *S. mutans* [25]. Our results also indicated that LF-1 could downregulate *S. mutans* LDH expression at the transcriptional and enzymatic levels. Therefore, the deficiency of *S. mutans* LDH and its acidogenicity induced by LF-1 signify a decrease in the cariogenic potential of *S. mutans*.

Furthermore, the acid tolerance of *S. mutans* can help it survive in a lethal acidified microenvironment [26], and membrane F-ATPase plays a vital role in maintaining pH homeostasis and determining the aciduricity of *S. mutans* [27]. Low pH activates the transcription and translation of *S. mutans* F-ATPase. As the primary acid tolerance mechanism, F-ATPase excretes excess protons in an energy-dependent manner to maintain intracellular pH stability [28]. LF-1 effectively inhibited the F-ATPase activity of *S. mutans* and downregulated the expression of the related gene *atpD*, leading to an acid tolerance deficiency and succumbing to this lethal acidic environment.

*S. mutans* possesses a strong ability for biofilm formation, where excess acid metabolites accumulate locally, leading to tooth demineralisation and dental caries. Early colonisation is the first step in *S. mutans’* biofilm formation [29]. Considering that the doubling time of *S. mutans* in the logarithmic phase is approximately 1.5 h [30], we confirmed that LF-1 could significantly weaken *S. mutans’* adhesion within 60 min. The inhibition of *S. mutans* adhesion by LF-1 may be attributed to reduced bacterial activity and the restriction of adhesion-related systems, such as GTFs [13] and VicRK [31]. Decreased *S. mutans* adhesion facilitates its maintenance in a planktonic state, contributing to the clearance of *S. mutans* by saliva and further delaying *S. mutans* aggregation and cariogenic biofilm formation [4].

EPS synthesis is another important cariogenic virulence factor of *S. mutans*, the roles of which are closely attributed to the *S. mutans* GTF system [13,14]. *S. mutans* GTFs can metabolise ingested sucrose into glucans, among which water-insoluble glucans can form colloidal EPS as the main component of the biofilm extracellular matrix [32]. EPS can stabilise biofilms in a compact and rigid structure and further aggravate the acidic cariogenic microenvironment. Additionally, glucans can provide binding sites for *S. mutans* and other oral microorganisms to help them further colonise and aggregate [33]. LF-1 significantly reduced water-insoluble EPS content in *S. mutans* biofilms at higher concentrations (≥4 μmol/L), where LF-1 slashed the transcriptional expression of *gtfB* and *gtfC,* which are mainly associated with the synthesis of insoluble glucans. LF-1 at 2 μmol/L only downregulated the expression of *gtfD*, which primarily generates soluble glucans for *S. mutans’* energy metabolism [34]. Hence, drug treatment at this concentration did not reduce the biomass or insoluble EPS production of *S. mutans* biofilms. In general, our study corroborated that LF-1 can effectively inhibit *S. mutans* biofilm formation with decreased bacteria and extracellular matrix by morphological observations and is mainly attributed to the suppression of the growth, adhesion, and EPS synthesis of *S. mutans*.

Considering that LF-1 can serve as a strong environmental stressor for *S. mutans*, we subsequently examined the effects of LF-1 on the key genes in the signalling pathways to mount coordinated responses [35]. These regulatory systems, including VicRK, LiaSR, and ComDE regulators, coordinate the expression of *S. mutans’* cariogenic virulence. VicRK is an essential signal transduction system for *S. mutans* to cope with acid and oxidative stress and to facilitate competence [31]. VicRK, along with LiaSR, contributes to the surface adhesion and stress tolerance of *S. mutans* [36]. ComDE is another crucial regulatory system (quorum-sensing system) for *S. mutans* that aids in its competence with other oral microbes [37]. Increasing the production of mutacins can help *S. mutans* achieve competence excellence, and further activation of downstream ComRS is directly responsible for the competence activation of *S. mutans*. Due to the reduced expression of the associated vital genes (including *vicR*, *liaR*, *comD*, and *comE*), LF-1 seriously impacts the *S. mutans* regulatory systems related to stress response and competence, which may further aggravate the LF-1-induced decline in growth activity, biofilm formation ability, and cariogenic capacity of *S. mutans*.

It is worth noting that 2 μmol/L was a particularly interesting concentration in this study, which is as low as 1/4 MIC for *S. mutans*. LF-1 at this concentration can still inhibit the growth and cariogenic virulence factors (including acidogenicity, aciduricity, and adhesion) of *S. mutans*. Although 2 μmol/L of LF-1 did not inhibit *S. mutans* biofilm formation, it significantly reduced the resident, thickness, and acid production capacity of the *S. mutans* biofilm. Furthermore, LF-1 at this concentration can also affect the expression of genes related to *S. mutans* regulatory systems. Therefore, 2 μmol/L is a critical concentration for LF-1 to effectively inhibit the cariogenic capacity of *S. mutans*, which is remarkably low in the reported literature related to AMPs for caries prevention and confers LF-1 with promising clinical application prospects [18].

The limitations and outlook of this study should be addressed in future investigations. The inhibition of *S. mutans’* cariogenic capacity by LF-1 is extensive and complicated in terms of transcriptional expression, enzyme activity, and key metabolites. Therefore, multi-omics (including transcriptomics, proteomics, and metabolomics) analysis should be conducted to study the effects of LF-1 on the biological functions of *S. mutans* (especially cariogenic capacity) and to explore the key pathways and mechanisms [38]. Moreover, we demonstrated that LF-1 possesses favourable antibacterial and anticaries activities against *S. mutans*. Due to the complex structure and composition of natural dental biofilms, it is necessary to verify the suppression of the cariogenic capacity of *S. mutans* by LF-1 in a multi-strain biofilm model and in vivo animal experiments to evaluate the characteristics of microbial composition and species interactions under drug treatment [39]. Finally, LF-1 may act as a promising anticaries drug, and future studies should focus on evaluating its clinical applications [40]. We can further assess its drug properties (including anticaries effects, stability, and adverse effects) in in vitro and in vivo models and explore suitable modes of drug delivery (such as chewing gum, toothpaste, and mouthwash). Clinical trials would be useful to validate its clinical applicability for caries prevention.

## 4. Materials and Methods

### 4.1. Peptide Synthesis and Bacterial Cultivation

The antimicrobial peptide LF-1 (WKLLRKAWKLLRKA) was synthesised, identified, and purified to >95% as previously described [19]. The peptide was stored at −20 °C and dissolved in sterilized, distilled, and deionised water (DDW) before use. Two-fold serial concentration gradients, including the MIC (8 μmol/L) and sub-MICs (4 μmol/L and 2 μmol/L) of LF-1, were adopted in this study. Sterilised DDW was used as the negative control (untreated).

The tested bacterial strain, *S. mutans* UA 159, was obtained from our laboratory. *S. mutans* was anaerobically cultured in brain-heart infusion (BHI; BD, Franklin Lakes, NJ, USA) broth (85% N_2_, 10% H_2_, and 5% CO_2_) at 37 °C.

### 4.2. Cell Viability Assay

We assessed the effect of LF-1 on the viability of planktonic *S. mutans* using MTT (Invitrogen, Carlsbad, CA, USA) staining, as previously described [41]. In brief, bacterial culture was diluted in BHI to 1 × 10^6^ colony-forming units (CFU)/mL and treated with LF-1 anaerobically for 6 h at 37 °C. Bacterial suspensions were centrifuged at 4500× *g* for 5 min at 4 °C. The treated bacteria were stained with MTT (dissolved in phosphate buffered saline [PBS] to 0.5 mg/mL) under dark conditions for 2 h at 37 °C. Intracellular formazan crystals were eluted in dimethyl sulfoxide (DMSO; Sigma-Aldrich, St. Louis, MO, USA), and aliquots (100 μL) were measured at an absorbance of 540 nm (A_540_) on a microplate reader (EPOCH 2, BioTek, Winooski, VT, USA). The results are expressed as cell viability relative to the untreated control.

### 4.3. Determination of Acid Production Capacity for Planktonic S. mutans

#### 4.3.1. Lactic Acid Measurement

Mid-logarithmic *S. mutans* were centrifugally collected (4500× *g*, 5 min, and 4 °C), washed twice with PBS, and resuspended in buffered peptone water (BPW; Nissui, Tokyo, Japan; supplemented with 0.2% sucrose) to 5 × 10^8^ CFU/mL. The bacterial suspensions were treated with LF-1 anaerobically for 2 h at 37 °C. After recentrifugation (8000× *g*, 5 min, and 4 °C), the supernatants were withdrawn for lactic acid measurement using the Lactate Assay Kit (Sigma-Aldrich). The absorbance at 570 nm (A_570_) was measured on a microplate reader and converted to the lactic acid concentration (mmol/L) [42].

#### 4.3.2. Glycolytic pH Drop Assay

The centrifugally collected bacteria (4500× *g*, 5 min, and 4 °C) were washed twice with 1% glucose-buffered salt solution (containing 0.5 mmol/L K_3_PO_4_, 37.5 mmol/L KCl, and 1.25 mmol/L MgCl_2_; pH 6.5) and resuspended in this buffer at 5 × 10^8^ CFU/mL. The bacterial suspensions were treated with LF-1, whose pH was monitored at 15 min intervals within 90 min using a pH meter (PB-10, Sartorius, Goettingen, Germany) [43].

#### 4.3.3. LDH Activity Assay

Bacterial culture was diluted in BHI to approximately 10^8^ CFU/mL and treated with LF-1 anaerobically for 1 h at 37 °C. The bacterial suspensions were centrifuged (8000× *g*, 5 min, and 4 °C), resuspended in lysis buffer, and sonicated on ice for 30 cycles of 3 s ultrasonic disruption in 10 s intervals. The lysate was centrifuged, and the supernatants were processed using the LDH Activity Assay Kit (Sigma-Aldrich). The absorbance at 450 nm was measured on a microplate reader to quantify the LDH activity (U/CFU) [44].

### 4.4. Determination of Acid Tolerance Capacity for Planktonic S. mutans

#### 4.4.1. Acid Tolerance Assay

We chose a pH of 5.0 as the lethal acidic environment and adjusted the tryptone-yeast extract-glucose (TYEG, containing 20 mmol/L glucose; BD) broth to pH 5.0, using a 40 mmol/L phosphate/citrate buffer. Mid-logarithmic *S. mutans* were diluted in TYEG to 1× 10^7^ CFU/mL and treated with LF-1 anaerobically for 2 h at 37 °C. The number of surviving bacteria in this lethal acidic environment was determined by counting the bacterial colonies [45].

#### 4.4.2. F-ATPase Activity Assay

Mid-logarithmic *S. mutans* were centrifuged (4500× *g*, 5 min, and 4 °C) and resuspended in 75 mmol/L Tris-HCl (containing 10 mmol/L MgSO_4_; pH 7.0). After adding toluene at a volume ratio of 1/10, the bacterial suspensions were vigorously vortexed, incubated for 5 min at 37 °C, and subsequently permeabilised in two freeze-thaw cycles (liquid nitrogen/37 °C). The permeabilised bacteria were recentrifuged, resuspended, and stored at −80°C before use.

Permeabilised bacteria were treated with LF-1 for 15 min at room temperature. The treated bacteria were resuspended in 50 mM Tris-maleate buffer (containing 10 mmol/L MgSO_4_ and 5 mmol/L ATP (Sigma-Aldrich; pH 6.0) and incubated for 30 min at 37 °C. The released inorganic phosphate was determined using the reduced-molybdophosphate spectrometric method, reflecting F-ATPase activity. The results are expressed as relative enzymatic activity relative to the untreated control [46].

### 4.5. Initial Bacterial Colonisation Assay

Bacterial cultures were diluted in BHI-sucrose (BHIS; BHI contains 1% sucrose) broth to 1 × 10^7^ CFU/mL, treated with LF-1, and incubated in 24-well plates containing circular microscope coverslips for 30 and 60 min at 37 °C, respectively. Circular microscope coverslips were pre-coated with salivary pellicles, and saliva was collected from healthy, caries-free volunteers after obtaining informed consent and the ethics committee’s approval. Subsequently, the coverslips were washed twice with PBS, and the attached bacteria were transferred into fresh PBS using a sterile pipette tip. The resulting bacterial colonies were counted to determine the number of initially colonised bacteria [30].

### 4.6. S. mutans Biofilm Formation Assay

*S. mutans* biofilms were formed in 24-well plates pre-coated with a salivary pellicle, as described previously with some modifications [47]. Bacterial cultures were diluted in BHIS to 1 × 10^6^ CFU/mL and added with LF-1. The bacterial suspensions were shaken slowly (100 rpm) for 5 min at 37 °C to achieve uniform mixing and then incubated anaerobically at 37 °C for 24 h. The formed biofilms were washed twice with PBS, fixed with methanol for 15 min, and stained with 0.1% CV (Sigma-Aldrich) for 5 min. The bound dye was eluted with 33% glacial acetic acid, and biofilm biomass was determined by measuring the eluate at an absorbance of 595 nm on a microplate reader. The results are expressed as the relative biomass percentage compared to the untreated control.

### 4.7. Water-Insoluble EPS and Lactic Acid Measurement for S. mutans Biofilm

Water-insoluble EPS content was determined using the anthrone method [47]. The biofilms formed were washed twice with PBS, scraped from the plates, and vortexed to disperse in fresh PBS. The bacterial suspensions were centrifuged (8000× *g*, 5 min, and 4 °C) and washed with PBS in three cycles to remove water-soluble EPS. The precipitates were resuspended in 1 mol/L NaOH for a sufficient reaction time of 2 h. After re-centrifugation, 200 μL of the supernatant was incubated with 600 μL 0.1% anthrone (dissolved in 80% sulfuric acid) for 6 min at 95 °C. The absorbance at 625 nm was measured using a microplate reader and converted to a water-insoluble EPS concentration (mg/L).

The lactic acid measurement protocol was similar to that described above. The formed biofilms were washed twice with PBS and re-incubated in BPW anaerobically for 2 h at 37 °C. The supernatants were processed using the Lactate Assay Kit, and lactic acid production was determined by measuring the A_570_ nm using a microplate reader.

### 4.8. SEM Observations of S. mutans Biofilm

As described earlier, *S. mutans* biofilm was formed on circular microscope coverslips pre-coated with a salivary pellicle. The formed biofilms were washed twice with PBS, fixed with 2.5% glutaraldehyde overnight at 4 °C, and dehydrated with increasing ethanol gradients (35%, 50%, 75%, 90% × 2, and 100% × 2) for 30 min in each solution. The dehydrated samples were dried to a critical point, gold-sprayed, and observed using SEM (FEI, Eindhoven, The Netherlands) [48].

### 4.9. CLSM Observations of S. mutans Biofilm

As described earlier, *S. mutans* biofilms were formed in glass-bottom dishes pre-coated with a salivary pellicle. The formed biofilms were stained using Syto-9 (Invitrogen) stain and a dextran fluorescent probe (Dextran, Alexa Fluor 647; Invitrogen) following the manufacturer’s instructions. Bacteria were stained with green fluorescence (Syto-9), and EPS was stained with red fluorescence (Alexa Fluor 647). The labelled biofilms were observed by CLSM (Zeiss, Oberkochen, Germany) at 40× magnification and three-dimensionally reconstructed using ZEN software (Zeiss). Biomass-related calculations were conducted using COMSTAT software (http://www.imageanalysis.dk; accessed on 26 February 2023) [49].

### 4.10. Real-Time Quantitative PCR (qPCR) of Virulence-Associated Genes

We performed qPCR to evaluate the effects of LF-1 on *S. mutans* virulence-associated gene expression. Bacterial culture was diluted in BHI broth to approximately 10^8^ CFU/mL and treated with LF-1 anaerobically for 1 h at 37 °C. According to the manufacturer’s instructions, bacterial RNA was extracted using the TRIzol Bacterial RNA Isolation Kit (Invitrogen), and cDNA was synthesised using a reverse transcription kit (RR047A; Takara, Shiga, Japan). Virulence-associated genes and their specific primers are listed in Table 2. The reaction mixture was prepared using cDNA and a qPCR kit (RR420A; Takara), and qPCR was performed using a Light Cycler 480 System (Roche, Basel, Switzerland). Gene expression was normalised to 16S rDNA gene transcription and was calculated using the 2^−ΔΔCt^ method [50].

### 4.11. Statistical Analysis

All data were obtained from at least three independent experiments and statistically analysed using GraphPad Prism software (version 9.4, GraphPad Software, San Diego, CA, USA) via one-way analysis of variance and Tukey’s honestly significant difference tests. Statistical significance was set at *p* < 0.05.

## 5. Conclusions

In conclusion, the null hypothesis was rejected, and the novel lactotransferrin-derived AMP LF-1 inhibited the growth and cariogenic virulence factors of *S. mutans in vitro*. LF-1 displayed powerful inhibition of the acidogenicity and aciduricity of *S. mutans* and reduced its key enzymatic activity. LF-1 also inhibited the adhesion and EPS synthesis of *S. mutans*, resulting in impaired biofilm formation. Furthermore, LF-1 downregulated *S. mutans* gene expressions related to cariogenic virulence and the stress response. Notably, 2 μmol/L is the critical concentration for LF-1 to inhibit the cariogenic capacity of *S. mutans* effectively. The remarkable inhibition of *S. mutans* growth and cariogenic capacity and the relatively low critical concentration confer LF-1 with promising applicability in caries prevention.

## Figures and Tables

**Figure 1 antibiotics-12-00563-f001:**
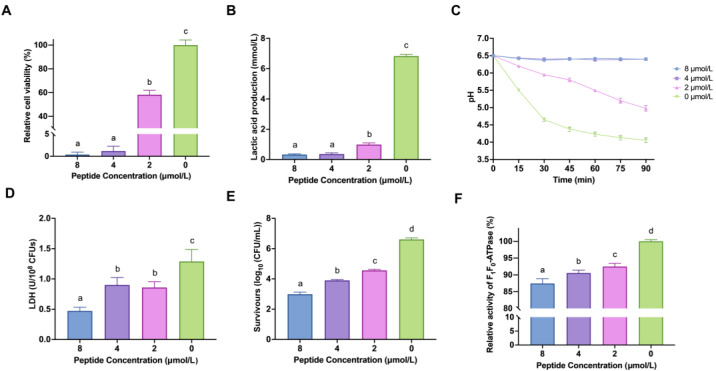
The inhibition of LF-1 on the growth, acid production, and acid tolerance of planktonic *Streptococcus mutans*. The relative cell viability of *S. mutans* under LF-1 treatment within 6 h was evaluated using the MTT method. (**A**) The acid production capacity of *S. mutans* under the LF-1 treatment is determined by measuring lactic acid production (**B**), glycolytic pH drop (**C**), and LDH activity. (**D**) The acid tolerance capacity of *S. mutans* under the LF-1 treatment is evaluated by counting survivors in the lethal acidic environment (**E**) and measuring the relative F_1_F_0_-ATPase activity. (**F**) All data are presented as the mean ± standard deviation from at least three independent experiments, and relative values are obtained by comparison with the untreated control. Columns labelled with different superscript letters denote significant statistical differences in one group (one-way analysis of variance; *p* < 0.05). MTT: methyl thiazolyl tetrazolium; LDH: lactate dehydrogenase.

**Figure 2 antibiotics-12-00563-f002:**
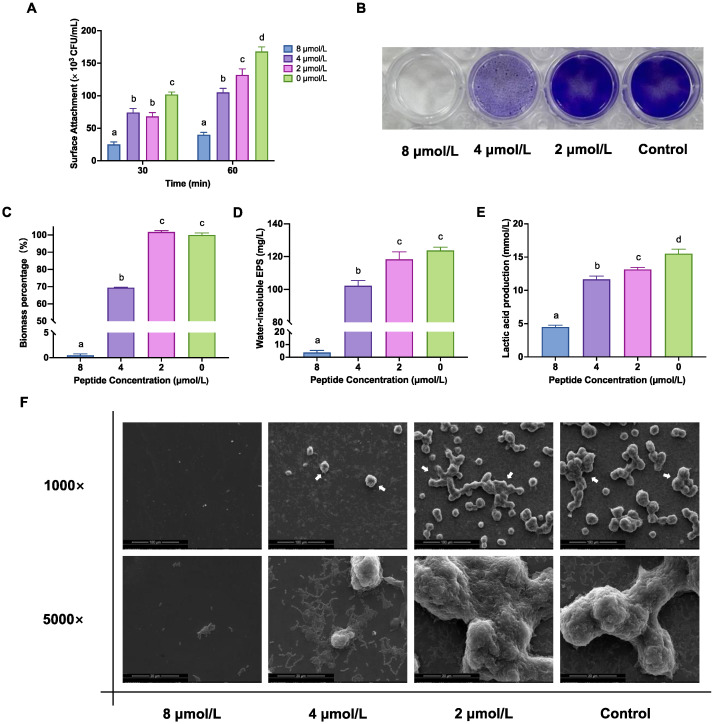
The inhibition of LF-1 on the adhesion and biofilm formation of *Streptococcus mutans*. LF-1 significantly decreases the surface-attached *S. mutans* in a concentration-dependent manner. (**A**) The inhibition of LF-1 on *S. mutans* biofilm formation is visually presented by the representative images of CV-stained biofilms (**B**) and assessed by measuring the biomass percentage (**C**), water-insoluble EPS (**D**), and lactic acid production. (**E**) SEM observations (**F**) further confirm that LF-1 alters the ultrastructural morphology of the *S. mutans* biofilm with visually reduced extracellular matrix and bacteria. The arrows indicate the bacterial clusters that comprise aggregated *S. mutans* wrapped in EPS. All data are presented as the mean ± standard deviation from at least three independent experiments, and relative values are obtained by comparison with the untreated control. Columns labelled with different superscript letters denote significant statistical differences in one group (one- or two-way analysis of variance; *p* < 0.05). CV: crystal violet; EPS: exopolysaccharide; SEM: scanning electron microscopy.

**Figure 3 antibiotics-12-00563-f003:**
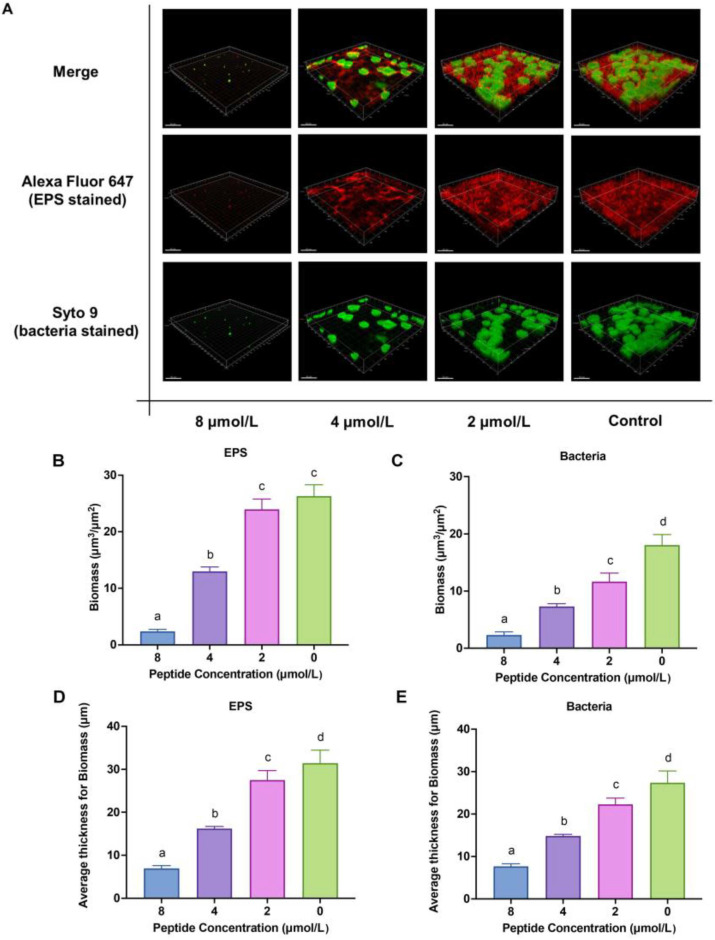
The three-dimensional structure of *Streptococcus mutans* biofilms under the LF-1 treatment for 24 h is observed by the CLSM, where EPS is stained red with Alexa Fluor 647 and bacteria are stained green with Syto 9. (**A**) Biomass (**B**,**C**) and average thickness (**D**,**E**) for EPS and bacteria are calculated according to five randomly selected images from the red and green channels, respectively. Data are presented as the mean ± standard deviation, and columns labelled with different superscript letters denote significant statistical differences in one group (one-way analysis of variance; *p* < 0.05). EPS: exopolysaccharide; CLSM: confocal laser scanning microscopy.

**Figure 4 antibiotics-12-00563-f004:**
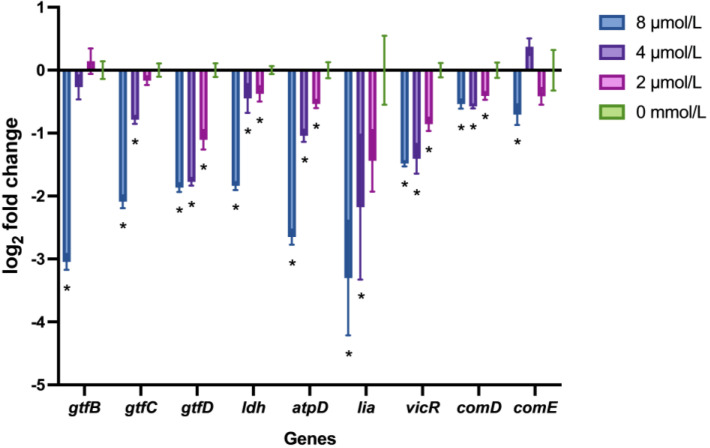
The expression profiles of the *Streptococcus mutans* virulence-associated genes under LF-1 treatment were evaluated by real-time quantitative PCR. The gene expressions are presented as log_2_ fold change (mean ± standard deviation) normalised to 16S rDNA gene transcription from at least three independent experiments. Asterisks (*) indicate a significant statistical difference from the untreated control (two-way analysis of variance; *p* < 0.05).

**Table 1 antibiotics-12-00563-t001:** A comparison between the main findings of LF-1 previously reported and those obtained in this study.

**The Main Findings Reported Previously** [19,20]	**The Main Findings Reported in This Study**
LF-1 showed antibacterial activity against *S. mutans* with MIC of 8 μmol/L and MBC of 16 μmol/L LF-1 showed rapid killing kinetics and eradicated *S. mutans* at 16 μmol/L within 60 min. LF-1 inhibited *S. mutans* biofilm formation with MBIC_90_ of 8 μmol/L. LF-1 reduced the bacterial content in 1-d preformed *S. mutans* biofilm starting at an effective concentration of 8 μmol/L. 32 μmol/L LF-1 treatment exerted significant anticaries effects in vivo. A higher concentration (64 μmol/L) of LF-1 induced intensively destructive effects on the membrane and intracellular structure of *S. mutans*. A lower concentration (16 μmol/L) of LF-1 triggered the S.mutans membrane to form a mesosome-like structure. LF-1 displayed selective antibacterial activity against *S. mutans* compared with other oral streptococci (*S. sanguinis* and *S. gordonii*). *S. mutans* exhibited an apparent capacity to adsorb LF-1. LF-1 induces stronger cell membrane disruption in *S. mutans* than in the other oral streptococci. LF-1 displayed favourable stability in the conventional solutions, including DDW, PBS, and HEPES. LF-1 displayed low haemolytic toxicity and cytotoxicity under 64 μmol/L. LF-1 displayed satisfactory biocompatibility in the rodent model.	LF-1 at 2 μmol/L interfered with the cell viability of planktonic *S. mutans* within 6 h. LF-1 inhibited the acidogenicity of planktonic *S. mutans* in a concentration-dependent manner, which reduced lactic acid production in *S. mutans,* delayed the glycolytic pH drop in the culture medium, and decreased the LDH activity of *S. mutans*. LF-1 inhibited the aciduricity of planktonic *S. mutans* in a concentration-dependent manner, which reduced the survival of *S. mutans* in a lethal acidic environment and suppressed the F-ATPase activity. The surface attached *S. mutans* was significantly reduced when treated with LF-1 for 30 and 60 min. LF-1 at 4 and 8 μmol/L inhibited *S. mutans* biofilm formation with reduced insoluble EPS synthesis. LF-1 at 2 μmol/L retard the lactic acid production in the *S. mutans* biofilm. LF-1 altered the ultrastructural morphology of the *S. mutans* biofilm with an impaired structure accompanied by a visually reduced extracellular matrix and bacteria. LF-1 downregulated the *S. mutans* virulence-associated gene expression. Remarkably, 2 μmol/L is a critical concentration for LF-1 to effectively inhibit the cariogenic capacity of *S. mutans.*

Abbreviations: MIC, minimum inhibitory concentration; MBC, minimum bactericidal concentration; MBIC90, the minimal peptide concentration to inhibit ≥ 90% biofilm formation; DDW, distilled deionised water; PBS, phosphate-buffered saline; HEPES, 4-(2-hydroxyethyl)-1-piperazineethanesulfonic acid; LDH, lactate dehydrogenase; EPS, exopolysaccharide.

**Table 2 antibiotics-12-00563-t002:** Specific primers for the *S. mutans* virulence-associated genes.

Genes	Primer Sequence (Forward and Reverse)
** *16S rDNA* **	F: AGCGTTGTCCGGATTTATTG R: CTACGCATTTCACCGCTACA
** *gtfB* **	F: CACTATCGGCGGTTACGAAT R: CAATTTGGAGCAAGTCAGCA
** *gtfC* **	F: GATGCTGCAAACTTCGAACA R: TATTGACGCTGCGTTTCTTG
** *gtfD* **	F: TTGACGGTGTTCGTGTTGAT R: AAAGCGATAGGCGCAGTTTA
** *ldh* **	F: AAAAACCAGGCGAAACTCGC R: CTGAACGCGCATCAACATCA
** *atpD* **	F: TGTTGATGGTCTGGGTGAAA R: TTTGACGGTCTCCGATAACC
** *lia* **	F: CATGAAGATTTAACAGCGCG R: CGTCCTGTGGCACTAAATGA
** *vicR* **	F: CGTGTAAAAGCGCATCTTCG R: AATGTTCACGCGTCATCACC
** *comD* **	F: TTCCTGCAAACTCGATCATATAGG R: TGCCAGTTCTGACTTGTTTAGGC
** *comE* **	F: TTCCTCTGATTGACCATTCTTCTG R: GAGTTTATGCCCCTCACTTTTCAG

## Data Availability

Not applicable.

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
