# Peer review of "Novel Lactotransferrin-Derived Antimicrobial Peptide LF-1 Inhibits the Cariogenic Virulence Factors of Streptococcus mutans"

_antibiotics, 2023, doi:10.3390/antibiotics12030563_

Round 1

Reviewer 1 Report

This manuscript reports experiments on S. mutans physiology as affected by an antimicrobial peptide (LF-1) previously investigated from other perspectives by this group.  The major points are that acid generation, biofilm formation, EPS production, and virulence-associated gene expression are suppressed by LF-1.  The manuscript is well written and the figures, while very tiny and difficult to read, are well designed. Perhaps a larger font would help.  My questions revolve around data interpretation with respect to the methodology employed.

Please clarify how the bacteria were transferred from coverslips to PBS prior to counting (section 4.5).  What does “using a sterile pipette tip” mean?  Were they scraped from the coverslip surface?  If this is the case, one might expect less variation at low biomass than at high biomass, simply because it is difficult to remove the biofilm completely.  This appears to be the case in Fig 2a when comparing the 8 µmol/L LF-1 treatment at 30 min vs at 60 min - time of attachment seems to make little difference in the number of cells on the coverslip.  However, the no-treatment control has a clear increase in cells on the coverslip at 60 min vs at 30 min.

In the biofilm formation assay (section 4.6), the bacteria were incubated for 24 hrs in the presence of the AMP.  Does LF-1 degrade over this incubation period, or is it adsorbed onto the cell surfaces?  Either of these would result in a change in AMP concentration over time in this long assay.  In terms of practical application, how would such long exposure times be attained in the oral cavity?

Regarding lactic acid production in the 24-hr-old biofilms, it seems odd that the concentration is only marginally greater (2x?) in the densely growing biofilms (Fig 2e) than in the short-term planktonic experiment (Fig 1b).  Are the lactate concentrations similar to those seen by other investigators using similar systems (i.e., for the control)?  If we agree the values seem a bit low for the biofilms (or a bit high for the short planktonic incubation), could you provide a hypothetical explanation?

In Fig 2f, I don’t see much difference in EPS between the treatments.  Clearly, fewer cells are present when exposed to LF-1, but a major difference in morphology is difficult to detect, especially at 1000x (please use a scale bar rather than fold-magnification).  I am wondering whether the EPS measurements (Fig 2d) simply reflect reduced biomass.  When EPS values (Fig 3b) are normalized to cell biomass (Fig 3c), the 4- and 2-µmol/L ratios are the same (at about 0.5), but they are different from the 0- and 8-µmol/L values (which are also the same at about 1).  Admittedly it is difficult to make this comparison based on the bar graphs, but that’s how it appears to me.  More exact numbers might clarify this.  

The gene expression data (Fig 4) are impressive.  These were normalized to 16S gene activity.  Is 16S activity stable during LF-1 treatment?  This more a question than a criticism.

Reviewer 2 Report

This is a comprehensive study and a very well written paper. The dose response studies tell a compelling story regarding the influence of the LF-1 AMP.

Only minor changes are needed.

Line 386. Specify whether or not agitation (shear force) was used with the biofilm models.

In the discussion or methods, mention the stability of the LF-1 AMP under the most adverse pH conditions used in the study (e.g. at pH 5.0) if this is known, otherwise flag the stability under low pH conditions as an issue to be explored in future.

In the references, remove capital letters at the start of each word, to align with the journal style (refs 2,6,10,11,15,25,28,29,31.36-39.41,47,48). 

Reviewer 3 Report

Very interesting paper.

Figures very useful for understanding the results. Unfortunately the figures are too small, the legends need to be better detailed (a,b,c, are not described), furthermore the statistical differences between groups are not very evident. All figures need to be edited.

 This study is a further completion of studies done by the authors previously (reported in the text of the paper and in reference). For better understanding, in the discussion, it is helpful to add a table. In this table a comparison can be made between that already reported in the previous papers and the new data. This is to highlight the usefulness of the new results.
